# Graph Inverse Reinforcement Learning
# from Diverse Videos

**Sateesh Kumar**    **Jonathan Zamora***    **Nicklas Hansen***
**Rishabh Jangir**    **Xiaolong Wang**
UC San Diego

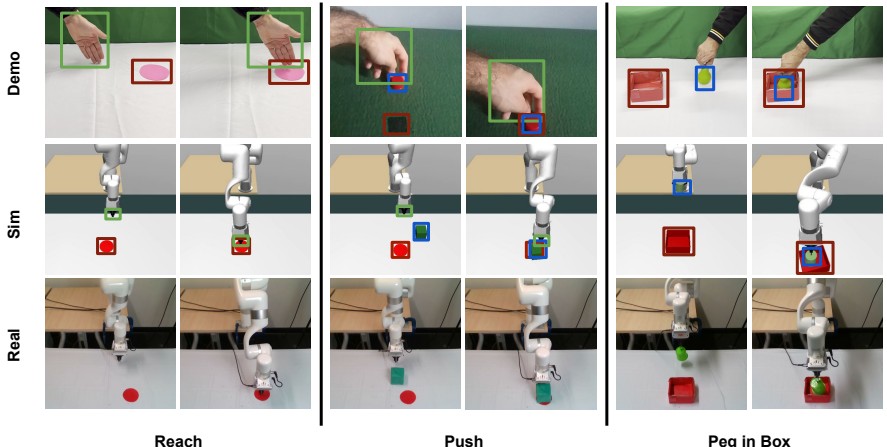

Figure 1: **GraphIRL.** We propose an approach for performing inverse reinforcement learning from diverse third-person videos via graph abstraction. Based on our learned reward functions, we successfully train image-based policies in simulation and deploy them on a real robot.

**Abstract:** Research on Inverse Reinforcement Learning (IRL) from third-person videos has shown encouraging results on removing the need for manual reward design for robotic tasks. However, most prior works are still limited by training from a relatively restricted domain of videos. In this paper, we argue that the true potential of third-person IRL lies in increasing the diversity of videos for better scaling. To learn a reward function from diverse videos, we propose to perform graph abstraction on the videos followed by temporal matching in the graph space to measure the task progress. Our insight is that a task can be described by entity interactions that form a graph, and this graph abstraction can help remove irrelevant information such as textures, resulting in more robust reward functions. We evaluate our approach, *GraphIRL*, on cross-embodiment learning in X-MAGICAL and learning from human demonstrations for real-robot manipulation. We show significant improvements in robustness to diverse video demonstrations over previous approaches, and even achieve better results than manual reward design on a real robot pushing task. Videos are available at https://sateeshkumar21.github.io/GraphIRL/.

**Keywords:** Inverse Reinforcement Learning, Third-Person Video, Graph Network

## 1   Introduction

Deep Reinforcement Learning (RL) is a powerful general-purpose framework for learning behavior policies from high-dimensional interaction data, and has led to a multitude of impressive feats in application areas such as game-playing [1] and robotics [2, 3]. Through interaction with an unknown environment, RL agents iteratively improve their policy by learning to maximize a reward signal, which has the potential to be used in lieu of hand-crafted control policies. However, the performance of policies learned by RL is found to be highly dependent on the careful specification of task-specific

6th Conference on Robot Learning (CoRL 2022), Auckland, New Zealand.

reward functions and, as a result, crafting a good reward function may require significant domain knowledge and technical expertise.

As an alternative to manual design of reward functions, *inverse RL* (IRL) has emerged as a promising paradigm for policy learning. By framing the reward specification as a learning problem, operators can specify a reward function based on video examples. While *imitation learning* typically requires demonstrations from a first-person perspective, IRL can in principle learn a reward function, *i.e.*, a measure of task progression, from *any* perspective, including third-person videos of humans performing a task. This has positive implications for data collection, since it is often far easier for humans to capture demonstrations in third-person.

Although IRL from third-person videos is appealing because of its perceived flexibility, learning a good reward function from raw video data comes with a variety of challenges. This is perhaps unsurprising, considering the visual and functional diversity that such data contains. For example, the task of pushing an object across a table may require different motions depending on the embodiment of the agent. A recent method for cross-embodiment IRL, dubbed XIRL [4], learns to capture task progression from videos in a self-supervised manner by enforcing temporal cycle-consistency constraints. While XIRL can in principle consume any video demonstration, we observe that its ability to learn task progression degrades substantially when the visual appearance of the video demonstrations do not match that of the target environment for RL. Therefore, it is natural to ask the question: *can we learn to imitate others from (a limited number of) diverse third-person videos?*

In this work, we demonstrate that it is indeed possible. Our key insight is that, while videos may be of great visual diversity, their underlying scene structure and agent-object interactions can be abstracted via a graph representation. Specifically, instead of directly using images, we extract object bounding boxes from each frame using an off-the-shelf detector, and construct a graph abstraction where each object is represented as a node in the graph. Often – in robotics tasks – the spatial location of an object by itself may not convey the full picture of the task at hand. For instance, to understand a task like *Peg in Box* (shown in Figure 1), we need to also take into account how the agent *interacts* with the object. Therefore, we propose to employ *Interaction Networks* [5] on our graph representation to explicitly model interactions between entities. To train our model, we follow [4, 6] and apply a temporal cycle consistency loss, which (in our framework) yields task-specific yet embodiment- and domain-agnostic feature representations.

We validate our method empirically on a set of simulated cross-domain cross-embodiment tasks from X-MAGICAL [4], as well as three vision-based robotic manipulation tasks. To do so, we collect a diverse set of demonstrations that vary in visual appearance, embodiment, object categories, and scene configuration; X-MAGICAL demonstrations are collected in simulation, whereas our manipulation demonstrations consist of real-world videos of humans performing tasks. We find our method to outperform a set of strong baselines when learning from visually diverse demonstrations, while simultaneously matching their performance in absence of diversity. Further, we demonstrate that vision-based policies trained with our learned reward perform tasks with greater precision than human-designed reward functions, and successfully transfer to a real robot setup with only approximate correspondence to the simulation environment. Thus, our proposed framework completes the cycle of learning rewards from real-world human demonstrations, learning a policy in simulation using learned rewards, and finally deployment of the learned policy on physical hardware.

## 2 Related Work

**Learning from demonstration.** Conventional imitation learning methods require access to expert demonstrations comprised of observations and corresponding ground-truth actions for every time step [7, 8, 9, 10], for which kinesthetic teaching or teleoperation are the primary modes of data collection in robotics. To scale up learning, video demonstrations are recorded with human operating the same gripper that the robot used, which also allows direct behaviro cloning [11, 12]. More recently, researchers have developed methods that instead infer actions from data via a learned forward [13] or inverse [14, 15] dynamics model. However, this approach still makes the implicit assumption that imitator and demonstrator share a common observation and action space, and are therefore not directly applicable to the cross-domain cross-embodiment problem setting that we consider.

**Inverse RL.** To address the aforementioned limitations, inverse RL has been proposed [16, 17, 18, 19, 20, 21] and it has recently emerged as a promising paradigm for cross-embodiment imitation in particular [22, 23, 24, 25, 26, 27, 28, 4, 29]. For example, Schmeckpeper et al. [22] proposes

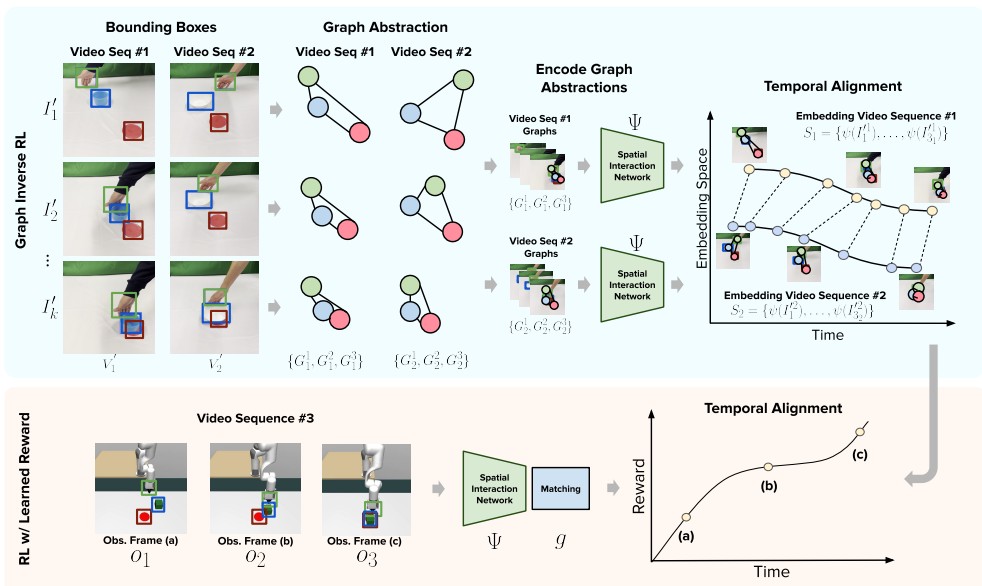

Figure 2: **Overview.** We extract object bounding boxes from video sequences using an off-the-shelf detector, and construct a graph abstraction of the scene. We model graph-abstracted object interactions using *Interaction Networks* [5], and learn a reward function by aligning video embeddings temporally. We then train image-based RL policies using our learned reward function, and deploy on a real robot.

a method for integrating video demonstrations without corresponding actions into off-policy RL algorithms via a latent inverse dynamics model and heuristic reward assignment, and Zakka et al. [4] (XIRL) learns a reward function from video demonstrations using temporal cycle-consistency and trains an RL agent to maximize the learned rewards. In practice, however, inverse RL methods such as XIRL are found to require limited visual diversity in demonstrations. Our work extends XIRL to the setting of diverse videos by introducing a graph abstraction that models agent-object and object-object interactions while still enforcing temporal cycle-consistency.

**Object-centric representations.** have been proposed in many forms at the intersection of computer vision and robotics. For example, object-centric scene graphs can be constructed for integrated task and motion planning [30, 31, 32], navigation [33, 34], relational inference [35, 36], dynamics modeling [5, 37, 38, 39, 40, 41], model predictive control [42, 43, 44], grasping [45, 46] or visual imitation learning [47, 48, 49]. Similar to our work, Sieb et al. [49] propose to abstract video demonstrations as object-centric graphs for the problem of single-video cross-embodiment imitation, and act by minimizing the difference between the demonstration graph and a graph constructed from observations captured at each step. As such, their method is limited to same-domain visual trajectory following, whereas we learn a general alignment function for cross-domain cross-embodiment imitation and leverage *Interaction Networks* [5] for modeling graph-abstracted spatial interactions rather than relying on heuristics.

## 3 Our Approach

In this section, we describe our main contribution, which is a self-supervised method for learning a visually invariant reward function directly from a set of diverse third-person video demonstrations via a graph abstraction. Our Graph Inverse Reinforcement Learning (GraphIRL) framework, shown in Figure 2, consists of building an object-centric graph abstraction of the video demonstrations and then learn an embedding space that captures task progression by exploiting the temporal cue in the videos. This embedding space is then used to construct a *domain invariant* and *embodiment invariant* reward function which can be used to train any standard reinforcement learning algorithm.

**Problem Formulation.** Given a task $T$, our approach takes a dataset of video demonstrations $D = \{V_1, V_2, \ldots, V_n\}$. Each video consists of image frames $\{I_1^i, I_2^i, \ldots, I_k^i\}$ where $i$ denotes the video frame index and $k$ denotes the total number of frames in $V_i$. Given $D$, our goal is to learn a reward function that can be used to solve the task $T$ for any robotic environment. Notably, we do *not*

assume access to any action information of the expert demonstrations, and our approach does *not* require objects or embodiments in the target environment to share appearance with demonstrations.

## 3.1 Representation Learning

To learn task-specific representations in a self-supervised manner, we take inspiration from Dwibedi et al. [6] and employ a temporal cycle consistency loss. However, instead of directly using images, we propose a novel object-centric graph representation, which allows us to learn an embedding space that not only captures task-specific features, but depends *solely* on the spatial configuration of objects and their interactions. We here detail each component of our approach to representation learning.

**Object-Centric Representation.** Given video frames $\{I_1^i, I_2^i, \ldots, I_k^i\}$, we first extract object bounding boxes from each frame using an off-the-shelf detector. Given $N$ bounding boxes for an image, we represent each bounding box as a $4 + m$ dimensional vector $o_j = \{x_1, y_1, x_2, y_2, d_1, d_2, \ldots, d_m\}$, where the first 4 dimensions represent the leftmost and rightmost corners of the bounding box, and the remaining $m$ dimensions encode distances between the centroids of the objects. For each frame $I_j^i$ we extract an object-centric representation $I_j'^i = \{o_1, o_2, \ldots, o_m\}$ such that we can represent our dataset of demonstrations as $D' = \{V_1', V_2', \ldots, V_n'\}$ where $V_i'$ is the sequence of bounding boxes corresponding to video $V_i$. Subsequent sections describe how we learn representations given $D'$.

**Spatial Interaction Encoder.** Taking inspiration from recent approaches on modeling physical object dynamics [5, 37], we propose a *Spatial Interaction Encoder Network* to explicitly model object-object interactions. Specifically, given a sequence $V_i'$ from $D'$, we model each element $I'$ as a graph, $G = (O, R)$, where $O$ is the set of objects $\{o_1, o_2, \ldots, o_m\}$, $m$ is the total number of objects in $I'$, and $R$ denotes the relationship between objects (*i.e.*, whether two objects interact with each other). For simplicity, all objects are connected with all other objects in the graph such that $R = \{(i, j) \mid i \neq j \land i \leq m \land j \leq m\}$. We use a fully-connected graph because this makes the least assumption about the problem and task specific object interaction structure could then be learned directly from the data. We compose an object embedding for each of $o_i \in O$ by combining *self* and *interactional* representations as follows:

$$f_o(o_i) = \phi_{\text{agg}}(f_s + f_{\text{in}}) \quad \text{with} \quad f_s(o_i) = \phi_s(o), \qquad f_{\text{in}}(o_i) = \sum_{j=1}^{m} \phi_{\text{in}}(o_i, o_j) \mid (i, j) \in \mathbb{R}, \quad (1)$$

where $f_s(o_i)$ represents the *self* or independent representation of an object, $f_{\text{in}}$ represents the *interactional* representation, *i.e.*, how it interacts with other objects in the scene, $f_o$ is the final object embedding, and $(,)$ represents concatenation. Here, the encoders $\phi_s$, $\phi_{\text{in}}$ and $\phi_{\text{agg}}$ denote Multi layer Perceptron (MLP) networks respectively. We emphasize that the expression for $f_{\text{in}}(\cdot)$ implies that the object embedding $f_o(.)$ depends on *all* other objects in the scene; this term allows us to model relationships of an object with the others. The final output from the spatial interaction encoder $\psi(\cdot)$ for object representation $I'$ is the mean of all object encodings:

$$\psi(I') = \frac{1}{m} \sum_{i}^{m} f(o_i). \qquad (2)$$

The spatial interaction encoder is then optimized using the temporal alignment loss introduced next.

**Temporal Alignment Loss**. Taking inspiration from prior works on video representatoin learning [6, 50, 51], we use temporal alignment as a proxy for video representation learning. Given a pair of videos, the task of self-supervised alignment implicitly assumes that there exists true semantic correspondence between the two sequences, *i.e.*, both videos share a common semantic space. These works have shown that optimizing for alignment leads to representations that could be used for tasks that require understanding task progression such as action-classification. This is because in order to solve for alignment, a learning model has to learn features that are (1) common across most videos and (2) exhibit temporal ordering. For a sufficiently large dataset with single task, the most common visual features would be distinct phases of a task that appear in all videos and if the task has small permutations, these distinct features would also exhibit temporal order. In such scenarios, the representations learned by optimizing for alignment are *task-specific* and invariant to changes in viewpoints, motions and embodiments.

In this work, we employ Temporal Cycle Consistency (TCC) [6] loss to learn temporal alignment. TCC optimizes for alignment by learning an embedding space that maximizes one-to-one nearest neighbour mappings between sequences. This is achieved through a loss that maximizes for cycle-consistent nearest neighbours given a pair of video sequences. In our case, the cycle consistency

is applied on the *graph abstraction* instead of image features as done in the aforementioned video alignment methods. Specifically, given $D'$, we sample a pair of bounding box sequences $V'_i = \{I'^i_1, \ldots, I'^i_{m_i}\}$ and $V'_j = \{I'j_1, \ldots, I'^j_{m_j}\}$, we extract embeddings by applying our spatial interaction encoder defined in Equation 2. Thus, we obtain the encoded features $S_i = \{\psi(I'^i_1), \ldots, \psi(I'^i_{m_i})\}$ and $S_j = \{\psi(I'^j_1), \ldots, \psi(I'^j_{m_j})\}$. For the $n$th element in $S_i$, we first compute its nearest neighbour, $v^n_{ij}$, in $S_j$ and then compute the probability that it cycles-back to $n$, $\beta^k_{ijn}$ as

$$ v^n_{ij} = \sum_k^{m_j} \alpha_k S^k_j \,, \alpha_k = \frac{e^{-||S^n_i - S^k_j||^2}}{\sum_k^{m_j} e^{-||S^n_i - S^k_j||^2}} \,, \beta^k_{ijn} = \frac{e^{-||v^n_{ij} - S^k_i||^2}}{\sum_k^{m_j} e^{-||v^n_{ij} - S^k_i||^2}} \,. \tag{3} $$

The cycle consistency loss for $n$th element can be computed as $L^{ij}_n = (\mu^n_{ij} - n)^2$, where $\mu^n_{ij} = \sum_k^{mi} \beta^k_{ijn} k$ is the expected value of frame index $n$ as we cycle back. The overall TCC loss is then defined by summing over all pairs of sequence embeddings $(S_i, S_j)$ in the data, *i.e.*, $L^n_{ij} = \sum_{ijn} L^n_{ij}$.

## 3.2 Reinforcement Learning

We learn a task-specific embedding space by optimizing for temporal alignment. In this section, we define how to go from this embedding space to a reward function that measures task progression. For constructing the reward function, we leverage the insight from Zakka et al. [4] that in a task-specific embedding space, we can use euclidean distance as a notion of task progression, *i.e.*, frames far apart in the embedding space will be far apart in terms of task progression and vice versa. We therefore choose to define our reward function as $r(o) = -\frac{1}{c}||\psi(o) - g||^2$, with $g = \sum_{i=1}^n \psi(I'^i_{m_i})$ , where $o$ is the current observation, $\psi$ is our Interaction Networks-based encoder from Section 3, $g$ is the representative goal frame, $m_i$ is the length of sequence $V'^i$ and $c$ is a scaling factor. This definition gives us a dense reward because as the observed state gets closer and closer to the goal, the reward starts going down and approaches zero when the goal and current observation are close in embedding space; refer to supplementary material for qualitative analysis of the learned reward function. After constructing the learned reward, we can use it to train any standard RL algorithm. We note that, unlike previous approaches [22, 4], our method does not use *any* environment reward to improve performance, and instead relies *solely* on the learned reward, which our experiments demonstrate is sufficient for solving diverse robotic manipulation tasks.

## 4 Experiments

In this section, we demonstrate how our approach uses diverse video demonstrations to learn a reward function that generalizes to unseen domains. In particular, we are interested in answering the questions: (1) How do vision-based methods for IRL perform when learning from demonstrations that exhibit *domain shift*? (2) Can our approach learn a stronger reward signal under this challenging setting? To that end, we first conduct experiments on the X-MAGICAL benchmark [4]. We then conduct experiments on multiple robot manipulation tasks using a diverse set of demonstrations.

**Implementation Details.** All MLPs have 2 hidden layers, and the embedding layer outputs features of size 128 in all experiments. We use Soft Actor-Critic (SAC) [52] as backbone RL algorithm for all methods. For experiments on X-MAGICAL, we follow Zakka et al. [4] and learn a state-based policy. For robotic manipulation experiments, we learn a multi-view image-based SAC policy [53]. For fair comparison, we only change the learned reward function across methods and keep the RL setup identical. Refer to the supplementary material for further implementation details.

**Baselines.** We compare against multiple vision-based approaches that learn rewards in a self-supervised manner: **(1) XIRL** [4] that learns a reward function by applying the TCC [6] loss on demonstration video sequences, **(2) TCN** [54] which is a self-supervised contrastive method for video representation learning that optimizes for temporally disentangled representations, and **(3) LIFS** [55] that learns an invariant feature space using a dynamic time warping-based contrastive loss. Lastly, we also compare against the manually designed **(4) Environment Reward** from Jangir et al. [53]. The environment reward baseline is an oracle method since it is a dense reward and is carefully designed for the task under consideration. For learning-based baselines, we use a ResNet-18 encoder pretrained on ImageNet [56] classification. See supplementary material for details.

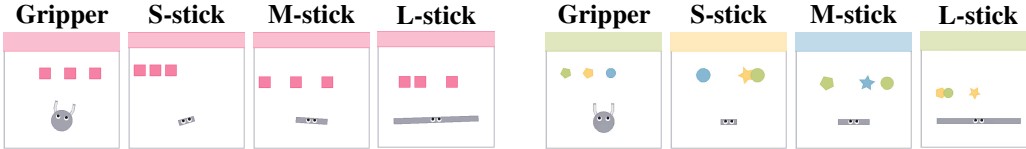

**Figure 3: Overview of X-MAGICAL task variants.** We consider two environment variants and four embodiments for our simulated sweeping task experiments. We evaluate IRL algorithms in both the *Diverse* and *Standard* environments across four embodiments in the *Cross-Embodiment settings*.

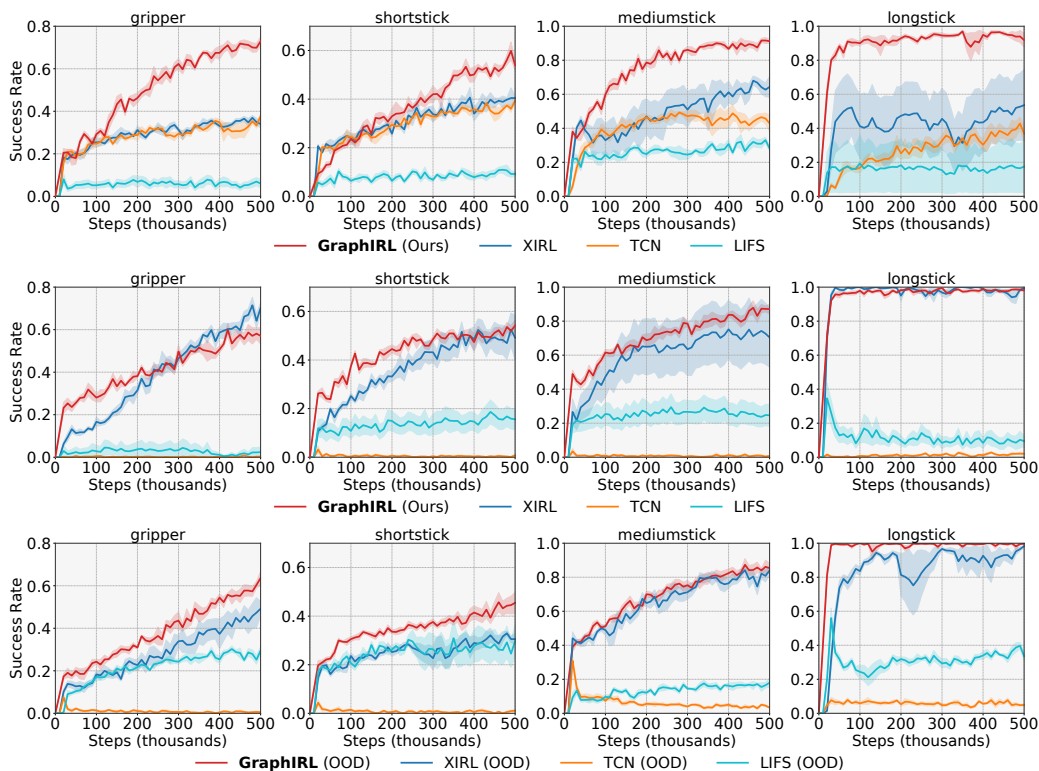

**Figure 4: Cross-Embodiment Cross-Environment.** Success rates of our method *GraphIRL* and baselines on *(top)* Standard Environment Pretraining → Diverse Environment RL, *(middle)* Diverse Environment Pretraining → Standard Environment RL and *(bottom)* Diverse (OOD) Environment Pretraining → Standard Environment RL. All reported numbers are averaged over 5 seeds. Our approach performs favorably when compared to other baselines on all three settings.

## 4.1 Experimental Setup

We conduct experiments under two settings: the *Sweep-to-Goal* task from X-MAGICAL [4], and robotic manipulation tasks with an xArm robot both in simulation and on a real robot setup. We describe our experimental setup under these two settings in the following.

**X-MAGICAL.** We choose to extend X-MAGICAL [4], a 2D simulation environment for cross-embodiment imitation learning. On this benchmark, we consider a multi-object sweeping task, where the agent must push three objects towards a static goal region. We utilize two variants of the X-MAGICAL benchmark, which we denote as *Standard* (original) and *Diverse* (ours) environments, shown in Figure 3. *Standard* only randomizes the position of objects, whereas *Diverse* also randomizes visual appearance. We consider a set of four unique embodiments {*gripper, short-stick, medium-stick, long-stick*}. In particular, we conduct experiments in the *cross-environment* and *cross-embodiment setting* where we learn a reward function in the *Standard* environment on 3 held-out embodiments and do RL in the *Diverse* environment on 1 target embodiment, or vice-versa. This provides an additional layer of difficulty for the RL agent as visual randomizations show the brittleness of vision-based IRL methods.

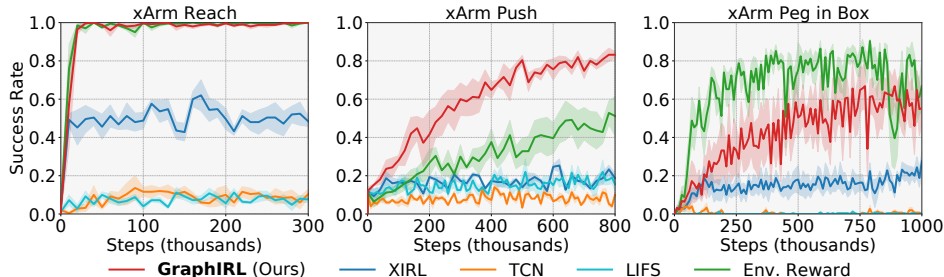

Figure 5: **Robotic Manipulation.** Success rates of our method *GraphIRL* and baselines on the tasks of *Reach*, *Push* and *Peg in Box*. All results are averaged over 5 seeds. We observe significant gains in performance specially over vision-based baselines due to large *domain-gap*

| *Real* | XIRL | Env. Reward | GraphIRL (Ours) |
|---|---|---|---|
| Push | 0.27 | 0.47 | 0.60 |
| Reach | 0.26 | 0.93 | 0.86 |
| Peg in Box | 0.06 | 0.60 | 0.53 |

Table 1: **Real robot experiments.** Success rate on robot manipulation tasks on physical hardware. We evaluate each method for 15 trials using a fixed set of goal and start state configurations.

**Robotic Manipulation.** Figure 1 shows initial and success configurations for each of the three task that we consider: **(1) Reach** in which the agent needs to reach a goal (red disc) with its end-effector, **(2) Push** in which the goal is to push a cube to a goal position, and **(3) Peg in Box** where the goal is to put a peg tied to the robot's end-effector inside a box. The last task is particularly difficult because it requires geometric 3D understanding of the objects. We collect a total of 256 and 162 video demonstrations for *Reach* and *Peg in Box*, respectively, and use 198 videos provided from Schmeckpeper et al. [22] for *Push*. The videos consist of human actors performing the same tasks but with a number of diverse objects and goal markers, as well as varied positions of objects. Unlike the data collected by Schmeckpeper et al. [22], we do not fix the goal position in our demonstrations. Additionally, we do not require the demonstrations to resemble the robotic environment in terms of appearance or distribution of goal location. In order to detect objects in our training demonstrations, we use a trained model from Shan et al. [57]. Refer to supplementary for details on robot platform.

## 4.2 Results

**X-MAGICAL.** Results for the *cross-embodiment and cross-environment* setting are shown in Figure 4. When trained on *Standard*, our method performs significantly better than vision-based baselines (*e.g.*, 0.58 GraphIRL for gripper vs 0.35 for XIRL and 0.99 GraphIRL for longstick vs 0.56 XIRL). We conjecture that vision-based baselines struggle with visual variations in the environment, which our method is unaffected by due to its graph abstraction. Additionally, we note that XIRL performs strongly when trained on *Diverse*. This could be because the *Diverse* environment contains some examples that are visually the same as *Standard* environment. To verify this, we conduct an experiment where we remove the shape (Square) and color (Red) used in *Standard* environment from the random configurations for *Diverse* environment and then construct a new set of demonstration. We refer to this environment as *Diverse (Out Of Distribution)*. In this setting, XIRL's performance drops significantly for 3 out of 4 embodiments.

**Robotic manipulation in simulation.** In this section, we answer the core question of our work: *can we learn to imitate others from diverse third-person videos?* In particular, we collect human demonstrations for manipulation tasks as explained in Section 4.1 and learn a reward function as explained in Section 3. This is a challenging setting because as shown in Figure 1, the collected data and robotic environments belong to different domains and do not share any appearance characteristics. Further, unlike previous works [22, 4], we do not use any environment reward as an additional supervision to the reinforcement learning agent. Figure 5 presents our results. For the **Reach** task, GraphIRL and environment reward achieve a success rate of 1.0, while other baseline methods are substantially behind GraphIRL (e.g. 0.477 XIRL and 0.155 TCN). In the **Push** setting, vision-based baseline methods still perform poorly. Similar to **Reach**, XIRL performed the best out of the vision-based baselines with a success rate of 0.187, and GraphIRL performed better than environment

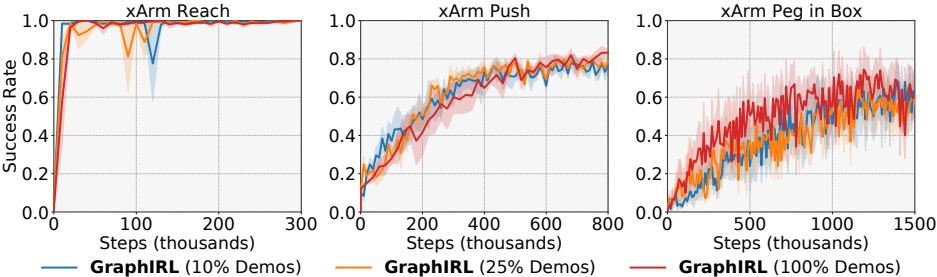

Figure 6: **Ablation Study on Number of Demonstrations.** Results averaged over 5 seeds. We use a total of $256$, $198$ and $162$ videos for *Reach*, *Push* and *Peg in Box* respectively.

reward with a success rate of $0.832$. The **Peg in Box** task is rigorous to solve since it requires 3-d reasoning and a precise reward function. Here, while all vision-based methods fail, GraphIRL solves the task with $55.2\%$ success rate. Overall, our GraphIRL method is able to solve 2D and 3D reasoning tasks with a real-robot without a hand-designed reward function or access to 3D scene information.

**Real robot experiments.** Finally, we deploy the learned policies on a real robot. For each experiment, we conduct 15 trials per method and report the average success rate. Results are shown in Table 1. Interestingly, we find that GraphIRL outperforms XIRL in all three tasks on the real robot setup, and on *Push*, GraphIRL performs better than the environment reward specifically designed for the task (e.g. $0.93$ Environment Reward vs $0.86$ GraphIRL) which is in line with our findings in simulation.

### 4.3  Ablations

**Impact of Modelling Spatial Interactions.** We study the impact of modeling object-object spatial interactions using Spatial Interaction Encoder Network (IN) described in Section 3.1. Specifically, we replace our proposed encoder component with an Multi-Layer Perceptron (MLP) by concatenating representations of all objects into a single vector and then feeding it to a 3-layer MLP network. Results in Table 2. We observe that modeling object interactions leads to a $20\%$ improvement in the RL success rate (i.e. $0.61$ for MLP vs $0.804$ for IN).

| Variant | Success Rate |
|---------|--------------|
| MLP | $0.61_{\pm 0.116}$ |
| IN | $0.804_{\pm 0.054}$ |

Table 2: Impact of modelling object-object interaction on *Push* task. **MLP:** Multi-layer perceptron and **IN:** Spatial Interaction Network Encoder. Results over 5 seeds.

**Impact of Decreasing Number of Demonstration Videos.** Results in Figure 6. We find that our approach is very data efficient and can learn meaningful rewards even from a small number of videos. It achieves decent RL success rate with only $10\%$ of total videos used (e.g. $72\%$ for Push).

## 5  Conclusions and Limitations

**Conclusion.** We demonstrate the effectiveness of our proposed method, *GraphIRL*, in a number of IRL settings with diverse third-person demonstrations. In particular, we show that our method successfully learns reward functions from human demonstrations with diverse objects and scene configurations, that we are able to train image-based policies in simulation using our learned rewards, and that policies trained with our learned rewards are more successful than both prior work and manually designed reward functions on a real robot. We also hereby commit to release our complete code and data to the public.

**Limitations.** While our method relaxes the requirements for human demonstrations, collecting the demonstrations still requires human labor; and although our results indicate that we can learn from relatively few videos, eliminating human labor entirely remains an open problem. Moreover, our approach assumes access to object bounding boxes. This implies that our method's performance is dependent on the performance of the object detector. Fortunately, 2d object detectors have become very reliable as we show in our experiments, we are able to use an off-the-shelf object detector to extract the bounding boxes without having to perform any manual labeling. Finally, the proposed graph abstraction allows us to solve tasks despite the large domain gap but it has some potential disadvantages too. In particular, we lose fine grained information such as object poses and precise object interactions which could be useful for complex tasks such as medical procedures. However, we conjecture that 2D images might also not be sufficient for such a task since inferring accurate object pose, 3D geometric information from a 2D image itself is a challenging problem.

**Acknowledgments**

This work was supported, in part, by grants from NSF CCF-2112665 (TILOS), NSF 1730158 CI-New: Cognitive Hardware and Software Ecosystem Community Infrastructure (CHASE-CI), NSF ACI-1541349 CC*DNI Pacific Research Platform, and gifts from Meta, Google, Qualcomm.

The authors would like to thank Sanjay Haresh, Kevin Zakka, Jianglong Ye and Mohit Jain for helpful discussions.

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
