# OpenReview forum: "Graph Inverse Reinforcement Learning from Diverse Videos"
_robot-learning.org/CoRL/2022/Conference — CoRL 2022 Oral_

### Official Review · Reviewer_77Z4 · 2022-07-29

**Originality:** Good
**Technical Quality:** Good
**Clarity Of Presentation:** Very Good
**Impact:** 3

**Recommendation:**

Weak Accept: I recommend accepting the paper, but will not argue for my recommendation if the majority of other reviewers have a different opinion.

**Summary:**

The authors present a technique they call GraphIRL, for learning reward abstractions across domains with diverse embodiment gaps. The method decomposes the underlying scene structure and agent-object interaction into a graph, which the authors claim is invariant to distractor features (like texture, agent embodiment, etc). The learned reward is then used in an IRL setting to learn policies from third-person demonstrations and they show successful cross-embodiment transfer on both simulated and real-world setups.

**Issues:**

Issues listed in the summary above.

**Quality Of The Limitations Section:**

Limitations are not well addressed

**Reviewer Expertise:**

5: The reviewer is absolutely certain that the evaluation is correct and very familiar with the relevant literature

**Robotics Focus:**

Sufficient demonstration on hardware

**Strengths And Weaknesses:**

Strengths

- The paper is highly polished and easy to read. The videos are high quality and the experimental results are quite thorough (with some details missing… see below).

- Real robot experiments are appreciated.

- This is an incredibly challenging problem domain so any improvement in performance could be impactful for the community.

Weaknesses

- Some references are missing. There have been quite a few papers that use object-centric representations to define (by definition) invariant state representations. One example would be Manuelli., “Keypoints into the Future: Self-Supervised Correspondence in Model-Based Reinforcement Learning.” I suggest the authors perform an additional literature search.

- The reward is not conditioned on policy actions and so implies some limitations in the form of reward that can be learned. e.g. learned rewards cannot be directly action dependent - or assumes the learned reward can infer the policy action if this is what is needed (concrete example: rewards that introduce penalties or costs for certain actions). Please add details discussing the fundamental limitations of the assertions made, especially since you have a “Limitations” section.

- Similarly, the reward is defined on a “representative goal frame” which also limits the rewards that can be learned (effectively, you can only represent rewards where some individual goal state is sufficient to solve the task). This should be discussed to avoid overselling the method.

- I’ve read the “Our Approach” section a few times and while I think I understand how the graph is constructed, a visual aid would be immensely helpful. I understand the component parts (psi, f_o, f_s, f_in), and I can follow these into S_i and S_j, but it would be nice to see the flow from image to bounding box, to embeddings to construction of the graph. Figure 2 is a good start, but doesn’t actually contain any of the terms from the “Our Approach” section so is somewhat hard to follow.

- Table 1: please explain why the environment reward underperforms your method? Is this a sparse reward (and if so why did you compare a dense reward with a sparse one)? Otherwise, I’d expect a well crafted true reward to be an upper bound so this requires some explanation.

- For the real robot experiments, did you train on any real data? Or did you train the reward only in sim and then deploy in real? Likewise, did you train the policy in real (or reward + policy trained in sim and deployed in real)? Sorry if this is spelled out, but I wasn’t able to confirm. If it’s the case that the reward was trained in sim only, it would make sense to me that a method that first crops objects learns a reward function that can better transfer sim to real, while a method like XIRL, that uses full image embeddings, won't perform as well (since background distractor features are going to result in generalization issues). This isn’t necessarily a ding against the method at all (and in fact using bounding boxes is a reasonable contribution), but an experiment that also trains the full stack in real would be necessary to make sure the graph representation accounts for the claimed performance gains (and it’s not just because of cropping out of the background).

- Nit: It would be helpful if the number of videos was in the Table 3 caption.

**Summary Of Recommendation:**

I’m on the fence between weak reject and weak accept. The proposed method is sound, however I have some questions pertaining to the real-robot experiments so that readers can better understand the performance implications of the designed reward. Likewise, an honest and comprehensive review of the method's limitations is needed (the discussion around requiring human demonstrations seems like a throwaway statement), however this seems easily addressable and go a long way to help frame the paper within the context of solutions to the problem.

Update: Bumping up to weak accept after rebuttal. The authors did a great job addressing some of my concerns in the updated version.

---

> ### Author Response · Authors · 2022-08-26
> **Authors' response to Reviewer 77Z4 [part 1/2]**
>
> We thank the reviewer for their thoughtful comments. Below, we address the individual comments:
>
> **Q**: Some references are missing. There have been quite a few papers that use object-centric representations to define (by definition) invariant state representations. One example would be Manuelli., “Keypoints into the Future: Self-Supervised Correspondence in Model-Based Reinforcement Learning.” I suggest the authors perform an additional literature search
>
> **A**: Thank you for pointing out the missing reference. We will add and discuss this work and relevant work in our related work. These works are related to our work on learning invariant representations, and different from our work since they use the representation as input for the forward dynamics model and we use the graph to learn and provide rewards as supervision signals.
>
> **Q**:  The reward is not conditioned on policy actions and so implies some limitations in the form of reward that can be learned. e.g. learned rewards cannot be directly action dependent - or assumes the learned reward can infer the policy action if this is what is needed (concrete example: rewards that introduce penalties or costs for certain actions). Please add details discussing the fundamental limitations of the assertions made, especially since you have a “Limitations” section.
>
> **A**: We agree. The form or our reward function doesn’t allow us to directly introduce penalties for certain actions. We will acknowledge this concern in the limitations section of our revision. Having said that, if the policy performs actions which lead to anomalous states far from the ones observed in the data, we expect to see a large distance between such a state and the goal state. This would translate to a large penalty by definition of our reward. This hypothesis follows directly from the anomaly detection experiments conducted in [6].
>
> **Q**: Similarly, the reward is defined on a “representative goal frame” which also limits the rewards that can be learned (effectively, you can only represent rewards where some individual goal state is sufficient to solve the task). This should be discussed to avoid overselling the method.
>
> **A**: We agree that for cases in which the goal state gives no indication about the completion of a task, our approach may not work. We will note this in the limitation section of our revision. However, most manipulation / robotic tasks such as those considered in this paper do have well defined goal state(s) and can be solved using the proposed approach.
>
> **Q**: I’ve read the “Our Approach” section a few times and while I think I understand how the graph is constructed, a visual aid would be immensely helpful. [..]. Figure 2 is a good start, but doesn’t actually contain any of the terms from the “Our Approach” section so is somewhat hard to follow.
>
> **A**: Thanks for pointing that out. We will improve Figure 2 and make it consistent with the terms defined in the approach section.
>
> **Q**: Table 1: please explain why the environment reward underperforms your method? Is this a sparse reward (and if so why did you compare a dense reward with a sparse one)? Otherwise, I’d expect a well crafted true reward to be an upper bound so this requires some explanation.
>
> **A**: This happens for the push task. The environment reward for push is a dense reward and it is the negative distance between the goal position and the current position of the object. We find that the learned reward gives a much higher reward when the object gets very close to the goal therefore the RL agent ends up learning a policy which strives more towards precision compared to the environment reward where the reward is linearly proportional to the distance. We find that with the environment reward, the RL policy aggressively pushes the object towards the goal in the beginning of the episode (with low precision).

---

> > ### Author Response · Authors · 2022-08-26
> > **Authors' response to Reviewer 77Z4 [part 2/2]**
> >
> > **Q**: For the real robot experiments, did you train on any real data? Or did you train the reward only in sim and then deploy in real? Likewise, did you train the policy in real (or reward + policy trained in sim and deployed in real)? Sorry if this is spelled out, but I wasn’t able to confirm
> >
> > **A**: The reward is trained only on the third-person demonstration data. There’s no reward training performed in simulation or on the real robot. The reward learned using the demonstrations is used to infer the reward during simulation for training RL policies. The policy is trained in simulation only and deployed in real robot without any additional training.
> >
> > **Q**: If it’s the case that the reward was trained in sim only, it would make sense to me that a method that first crops objects learns a reward function that can better transfer sim to real, while a method like XIRL, that uses full image embeddings, won't perform as well [..]. This isn’t necessarily a ding against the method at all (and in fact using bounding boxes is a reasonable contribution), but an experiment that also trains the full stack in real would be necessary to make sure the graph representation accounts for the claimed performance gains (and it’s not just because of cropping out of the background).
> >
> > **A**: We do not train the reward during simulation or real deployment. We learn the reward from demonstrations before training RL agent. Further, we would like to emphasize that during RL training, the policy and input image are kept identical for both vision based baselines (e.g., XIRL) and GraphIRL, the only difference is how the reward is computed for supervising the RL. Note that we have already observed a large gain from GraphIRL against the baselines in simulation (Figure 5 in the paper). The same gain has been transferred to the real robot experiments, not due to better handling the sim-to-real gap, but just because GraphIRL is already much better in simulation.
> >
> > Training on the real-robot could offset generalization issues but it significantly increases the cost for training. While, it would be interesting to compare self-supervised reward learning approaches with training directly on real, this is beyond the scope of our work, since we are primarily interested in answering whether learning from graph abstractions could improve generalization despite the significant domain gap between demonstrations used for reward learning and the robotic environment where the RL is performed.
> >
> > **Q**: It would be helpful if the number of videos was in the Table 3 caption.
> >
> > **A**: Thanks for the suggestion, we will add the number of videos in the caption of Table 3 in our revision.

---

### Official Review · Reviewer_NhHS · 2022-07-31

**Originality:** Very Good
**Technical Quality:** Very Good
**Clarity Of Presentation:** Very Good
**Impact:** 4

**Recommendation:**

Strong Accept: I recommend accepting the paper and will argue for my recommendation even if other reviewers hold a different opinion.

**Summary:**

This work proposes a framework for robot learning from third-person human demonstration videos. To learn from a diverse set of videos that have different viewpoints, motions, and embodiments, the IRL is performed at an abstract, graph level. They show that the reward function, indicating task progression, that is learned with this abstraction can be used to train an RL agent and the learned policy can be successfully transferred to a real robot.

**Issues:**

- How is occlusion handled? Do we always assume that all objects are visible?
- What is the action space for robotic tasks?
- Sim2real results: we need more information on what causes the performance drop here. What are some typical failure cases? Is that the policy? Perception? Action execution?
- Sim2real results: for Peg-in-Box, the performance in the real world is better than the one in simulation?
- Ln73-75: can you explain why approaches like BCO [14] assume shared observation / action space? At least not according to this review paper: Recent Advances in Imitation Learning from Observation (Torabi et al., 2019) see section 3.1.


**Quality Of The Limitations Section:**

Limitations are not well addressed

**Reviewer Expertise:**

3: The reviewer is fairly confident that the evaluation is correct

**Robotics Focus:**

Sufficient demonstration on hardware

**Strengths And Weaknesses:**

Plus
- The research problem is well motivated. The components in the proposed pipeline are designed well. There are no major flaws in experiments and evaluations.
- Choosing the right representation for an IRL problem is a fundamental research question. Scene graphs have a lot of potential in this direction and this work has demonstrated its effectiveness.
- No environment nor additional hand-crafted reward is required, which sets this paper apart from some previous works.

Minus
- This work assumes an object detector is given, which is a reasonable assumption. But there are several unanswered questions. What off-the-shelf detector is being used here? What’s the detection accuracy in the tasks presented? More critically, how accurate the perception module needs to be for the entire pipeline to be effective?
- Graph abstraction has all the advantages mentioned in the paper, but like any abstraction, some information is lost. What is some critical information that is lost in the abstraction? Does this prevent the proposed approach to generalize to tasks that heavily rely on, for example, object pose information, 3D geometric information, etc. This should be at least discussed in the limitations section.


**Summary Of Recommendation:**

Overall this is solid work. See strengths for the reasons for acceptance. Although the concerns raised should be addressed and discussed in detail.

---

> ### Author Response · Authors · 2022-08-26
> **Authors' response to Reviewer NhHS**
>
> We thank the reviewer for their thoughtful comments. Below, we address the individual comments:
>
> **Q**: This work assumes an object detector is given, which is a reasonable assumption. But there are several unanswered questions. What off-the-shelf detector is being used here? What’s the detection accuracy in the tasks presented? More critically, how accurate the perception module needs to be for the entire pipeline to be effective?
>
> **A**: We use [1] for detecting hands and objects in the scene. We find that our object detection framework is able to detect objects with 90-95% accuracy for all tasks. If an object isn’t detected, we use the detection from the previous frame. We assume all the objects would be visible at all times. We find that about 90% accurate object detection is sufficient for our approach. Performing object tracking might make the approach more robust to spurious detections but we have not applied it yet in our experiments. We will add more comprehensive details on object detection in the supplementary.
>
> [1] Understanding Human Hands in Contact at Internet Scale, Dandan Shan, Jiaqi Geng, Michelle Shu, David F. Fouhey, CVPR 2020.
>
>
> **Q**: Graph abstraction has all the advantages mentioned in the paper, but like any abstraction, some information is lost. What is some critical information that is lost in the abstraction? Does this prevent the proposed approach to generalize to tasks that heavily rely on, for example, object pose information, 3D geometric information, etc. This should be at least discussed in the limitations section
>
> **A**: We agree with the reviewer, due to graph abstraction, we lose fine grained information such as object poses and precise object interactions which could be useful for complex tasks such as medical procedures. However, we conjecture that 2D images might also not be sufficient for such a task since inferring accurate object pose, 3D geometric information from a 2D image itself is a challenging problem. We will add the discussed limitations in our revision.
>
> **Issues**
>
> **Q**: How is occlusion handled? Do we always assume that all objects are visible?
>
> **A**: We assume all objects are visible or partially visible. However, if an object isn’t detected by the object detection module, we use the detection from a previous frame.
>
> **Q**: What is the action space for robotic tasks?
>
> **A**: For Reach and Push, the action space is 2D (horizontal and vertical motion). Whereas for Peg in Box, the action space is 3D, where the 3rd degree of motion controls the height of the robotic arm.
>
>
> **Q**:  Sim2real results: we need more information on what causes the performance drop here. What are some typical failure cases? Is that the policy? Perception? Action execution?
> Sim2real results: for Peg-in-Box, the performance in the real world is better than the one in simulation?
>
> **A**: We note some discrepancy between the performance in real vs performance in sim. We find that our policy behavior changes when tested in the real world and this could be because our image based policy doesn’t generalize well from sim to real due to domain gap. Furthermore, we perform randomization in sim which could explain lower performance for peg-in-box. However, we note that the performance trend across methods remains same in both sim as well as real.
>
> We find that for the tasks of Push and Peg in Box, the robot fails to solve the task if the object / box is placed far from the robot.
>
>
> **Q**: Ln73-75: can you explain why approaches like BCO [14] assume shared observation / action space? At least not according to this review paper: Recent Advances in Imitation Learning from Observation (Torabi et al., 2019) see section 3.1.
>
> **A**: From Torabi et al., 2019, “The authors [Torabi et al., 2018], on the
> other hand, have proposed an algorithm, behavioral cloning from observation (BCO), that is instead concerned with learning generalized imitation policies using multiple demonstrations. The approach also learns an inverse dynamics model using an exploratory policy, and then uses that model to infer the actions from the demonstrations. Then, however, since the states and actions of the demonstrator are available, a regular imitation learning algorithm (behavioral cloning) is used to learn the task”
>
> The above description indicates that [14] is used to infer actions from demonstrations, once the actions are inferred, it reduces to a standard behavioral cloning algorithm and thus it would require demonstrations to share observation/action space.

---

### Official Review · Reviewer_A9wW · 2022-08-01

**Originality:** Very Good
**Technical Quality:** Fair
**Clarity Of Presentation:** Very Good
**Impact:** 4

**Recommendation:**

Weak Accept: I recommend accepting the paper, but will not argue for my recommendation if the majority of other reviewers have a different opinion.

**Summary:**

This paper proposes a method for learning a reward function which is both cross-embodiment and cross-domain. This allows for the reward function to be learned from human videos, for the learned reward to be used for learning in simulation, and for the learned policy to be deployed in the real world again on a physical robot. A graph representation is used to cross the domain gap, and temporal cycle consistency loss is used to cross the embodiment gap. Experiments show improved performance on a cross-embodiment benchmark, and successful deployment on a physical robot.

**Issues:**

1. Line 32: another important advantage of third-person videos, such as those used in this approach, compared to demonstrations involving a robot is the availability of in-the-wild data.
2. The description of TCC was quite concise, but it may be worth spending more space on this to explain it in more detail as it is quite key to the paper. The diagram from the original TCC paper conveys the intuition well.
3. In Eq (3), lowercase s is not defined, it seems. Probably it should be the same as uppercase S.
4. It may be helpful to emphasise that SAC is an oracle method in the real world experiments, in that it has access to reward function information

**Quality Of The Limitations Section:**

Limitations are not well addressed

**Reviewer Expertise:**

3: The reviewer is fairly confident that the evaluation is correct

**Robotics Focus:**

Sufficient demonstration on hardware

**Strengths And Weaknesses:**

Strengths:
1. The combination of graph representations and IRL appears to be novel and is likely to have significant impact, as this general approach can be used for many tasks.
1. The experimental finding that learned reward functions can be better than hand-designed rewards for these tasks is intriguing, and may be useful for the community.
1. The paper does well at proposing a vision for this learning pipeline: “learning rewards from real-world human demonstrations, learning a policy in simulation using learned rewards, and finally deployment of the learned policy on physical hardware”. This seems like a promising way forward for robot learning, and therefore raises the impact potential of this work.
1. Evaluation on a real robot shows that this method can be readily applied in the real world, supporting the vision for the full pipeline.
1. Data efficiency. As suggested by an ablation, the graph abstraction is much more data-efficient. A comparison using this metric to other methods may be interesting.

Limitations:
1. The node features only use the bounding box and centroids of each object, in the current draft of the paper. Other visual information is discarded. This is what helps the method cross the domain gap, but also drastically cuts the number of tasks that the method can be applied for, if it does not use visual features beyond the sizes of objects. For example, if the object to be pushed and the goal are the same size, then it is unclear how the network would be able to tell them apart.
1. X-MAGICAL is a useful benchmark, as it is designed specifically for evaluating cross-embodiment methods, but it is not very clear why strong performance in this benchmark would necessarily translate well to strong performance in real robot experiments. The policy is state-based rather than image-based. It seems like objects do not collide or interact much, so this might not show the full power of the Interaction Networks used in this method. The results on this benchmark suggest that when trained on not diverse videos but one visual style of videos, the proposed method generalises better to videos with different visual features. This makes sense because these visual features are all discarded by the representation, keeping only bounding boxes. However, in the second row of Figure 4, it seems like the baseline method XIRL performs just as well as the proposed method when the training dataset contains visually diverse videos, presumably because the visual encoder learns which features to focus on. This might lessen the impact of the current work, because it is also motivated by the context of learning from diverse videos.
1. The real-world experiments address a scenario which also might not show the full potential of Interaction Networks, as there are only 2-3 objects involved. The results are not very convincing for the proposed method unfortunately, in the current draft. The number of demonstrations is in the range of 162-256, which is high considering the simplicity of the task. For the peg-in-hole experiment, it does not seem as though 3D geometric understanding is needed to complete the task. In the video on the website, simply moving in a 2D line without raising peg allows peg to be dragged over wall and into box, due to compliance of string. In the demo videos, the peg is lifted over the wall -- it seems like the robot does not succeed in learning this behaviour. Additionally, it is not clear how the limited visual features in the graph representation would allow this 3D information to be captured. On Line 250, it is claimed that the proposed method outperforms the oracle method. This conclusion is not fully proven by the data as it stands. This is because (a) of the variance of the reward curves and (b) because this conclusion seems highly dependent on this specific termination point for the reward curve, when before that SAC was significantly out-performing GraphIRL, and after this point it is unclear which method will perform better, since neither curve looks like it has reached convergence.
1. The limitations section does not explore the limitations of the method in detail, such as those mentioned in the review.

Questions:
In Eq (4), why is g not just the final frame, but an aggregation across frames?


**Summary Of Recommendation:**

The paper proposes a novel idea and with high impact potential. However, there are some limitations introduced as a result of design decisions, and experimental evidence does not yet conclusively show the potential of this work. I will happily reconsider my weak reject recommendation if the visual features limitation is addressed and additional experiments demonstrate cleanly the advantages of this method.
- - -
EDIT: thank you to the authors for their hard work in completing the rebuttal. Taking the paper and discussion into consideration, I have now updated my recommendation to **Weak Accept**. The main reasons for this are as follows:
* The authors have clarified how 3D reasoning can still be performed by their method. Thank you for this: I now understand and agree.
* The authors have clarified that the object identity is included in the representation, albeit without visual features. With object identity and size, this is sufficient to solve some tasks already which are useful in the real world, and so it is sufficiently impactful as a method to merit acceptance. However, the fact remains that removing visual features is what allows this method to cross the sim2real gap, so extending this method to tasks which require the robot to use visual features will require significant changes, therefore this remains a limitation.
* The new experiment on OOD performance shows that the prior XIRL work does struggle with overcoming these visual distractor features, so the value of the proposed method has been made clearer.

Additional comments on the rebuttal:
* I appreciate the data efficiency experiment. This is a promising hypothesis to demonstrate an advantage of the graph representation. However, a comparison to other methods is required to demonstrate this advantage, since it is possible that the performance of other methods changes similarly when the training dataset size is changed. I understand that the authors have promised to run this later and include it in the final version if the paper is accepted, presumably due to time constraints.

Overall, graph representations seem like a useful and novel direction for IRL, and so this paper will be a valuable addition to CoRL 2022.

---

> ### Author Response · Authors · 2022-08-26
> **Authors' response to Reviewer A9wW [part 1/3]**
>
> We thank the reviewer for their thoughtful comments. Below, we address the individual comments:
>
> **Q**: The node features only use the bounding box and centroids of each object, in the current draft of the paper. Other visual information is discarded. This is what helps the method cross the domain gap, but also drastically cuts the number of tasks that the method can be applied for, [..]. For example, if the object to be pushed and the goal are the same size, then it is unclear how the network would be able to tell them apart.
>
> **A**: We agree that our current approach is mainly focusing on tasks considering geometric relations, and adding visual appearance features could potentially broaden its applications. However, it is non-trivial to utilize the visual appearance features. As shown in our experiments, the baseline with direct application on visual appearance features will fail with large domain gaps. Therefore, a more principled approach for fusing visual features into the proposed graph abstraction such that robustness to domain gap is preserved would be required. This is an exciting future direction but is beyond the scope of our current work.
>
> With that being said, our object detection pipeline also provides us the identity of the objects and thus our approach is able to distinguish between goal and object even if they are of the same size.
>
> **Q**: X-MAGICAL is a useful benchmark, [..], but it is not very clear why strong performance in this benchmark would necessarily translate well to strong performance in real robot experiments. [..]. However, in the second row of Figure 4, it seems like the baseline method XIRL performs just as well as the proposed method when the training dataset contains visually diverse videos, presumably because the visual encoder learns which features to focus on. This might lessen the impact of the current work, [..].
>
> **A**: One factor for XIRL’s improved performance when trained with diverse videos could be that the visual encoder learns to ignore visual appearance and starts focusing on the task as pointed out by the reviewer. However, we conjecture that another factor could be due to the way the diverse environment data is constructed. In particular, the diverse environment is constructed by introducing visual (shape and color) randomizations to the X-MAGICAL environment. Therefore, some example demonstrations could also have the shape and color that is used in the **Standard** environment. Thus, due to the presence of such examples, the encoder is able to learn representations that better solve the **Standard** environment. To verify this, we conduct an experiment where we remove the shape (Square) and color (Red)  used in **Standard** environment from the random configurations for **Diverse** environment and then construct a new set of demonstrations. This ensures that examples of **Standard** environment would indeed be out of distribution for the visual encoder.  We keep the number of examples the same as the experiments conducted in Figure 4.
>
> We find that for 3 out of 4 embodiments, there’s a significant decrease in performance for XIRL (e.g. 0.6722 vs 0.492 for Gripper) whereas the performance of GraphIRL remains relatively consistent under both settings. We report the final success rate for XIRL and GraphIRL below:
>
> | Embodiment    | Method   | Success Rate | Success Rate (Previous) |
> |---------------|----------|--------------|-------------------------|
> |     Gripper   |   XIRL   | 0.4920       | 0.6944                  |
> |               | GraphIRL | 0.6360       | 0.5707                  |
> |   Shortstick  |   XIRL   | 0.3053       | 0.5356                  |
> |               | GraphIRL | 0.4547       | 0.5413                  |
> |   Mediumstick |   XIRL   | 0.8400       | 0.76                    |
> |               | GraphIRL | 0.8533       | 0.8707                  |
> |   Longstick   |   XIRL   | 0.9827       | 0.9983                  |
> |               | GraphIRL | 0.9973       | 0.9813                  |
>
> In the table above, Success Rate (Previous) refers to the success rate that we had reported in the second row of Figure 4 where the diverse environment data also consisted of examples that belong to the standard environment.
>
> We will add this experiment in addition to the second row of Figure 4 in our paper. Please see the success rate curves under this setting [here](https://drive.google.com/file/d/1K_oLR8RO4hP9z8juV6Mab88wQ74F3kuC/view?usp=sharing).

---

> > ### Author Response · Authors · 2022-08-26
> > **Authors' response to Reviewer A9wW [part 2/3]**
> >
> > **Q**: The real-world experiments address a scenario which also might not show the full potential of Interaction Networks, as there are only 2-3 objects involved. The results are not very convincing for the proposed method unfortunately, in the current draft. The number of demonstrations is in the range of 162-256, which is high considering the simplicity of the task.
> >
> > **A**: We conduct experiments with 25% of the total demonstrations used for pre-training in our initial submission i.e. 64 videos for Reach, 49 videos for Push and 40 videos for Peg in Box we find that in all cases there’s only a small drop in performance (e.g. 83.2% vs 80.67% for Push). Further, we also conduct experiments with only 10% of demonstration data and find that GraphIRL gets decent performance for all tasks (e.g. 57.83% for Peg in Box and 72% for Push). This shows GraphIRL is very data efficient but adding more demonstrations generally leads to improved performance as expected. We will run other learning based baselines with reduced demonstrations and report the full results in the final version. We show the final success rate using GraphIRL for all tasks in simulation below:
> >
> > | Task       | % videos used | Success Rate |
> > |------------|---------------|--------------|
> > |     Reach  | 100%          | 1.0          |
> > |            | 25%           | 0.9967       |
> > |            | 10%           |        1.0      |
> > |    Push    | 100%          | 0.8320       |
> > |            | 25%           | 0.8067       |
> > |            | 10%           | 0.7200       |
> > | Peg in Box | 100%          | 0.6160       |
> > |            | 25%           | 0.5640       |
> > |            | 10%           | 0.5783       |
> >
> > Please see success rate curves for this setting [here](https://drive.google.com/file/d/1_8-Lp1WBnUNcJ0XcTdLBiL3VLCEF_BbS/view?usp=sharing).
> >
> >
> > **Q**: For the peg-in-hole experiment, it does not seem as though 3D geometric understanding is needed to complete the task. In the video on the website, simply moving in a 2D line without raising peg allows peg to be dragged over wall and into box, due to compliance of string. In the demo videos, the peg is lifted over the wall -- it seems like the robot does not succeed in learning this behaviour.
> >
> > **A**: We note that the policy learned via GraphIRL does actually lift the peg over the wall and insert it inside the box during simulation and exhibits a behavior similar to that learned using the environment reward. If the robot moves along a 2D line, the peg would be too high and won’t be dropped into the box. The failure to lift the peg completely over the wall during real deployment is more of an artifact due to sim2real gap.
> >
> >
> > **Q**: Additionally, it is not clear how the limited visual features in the graph representation would allow this 3D information to be captured.
> >
> > **A**: We agree the graph representation in GraphIRL doesn’t capture 3D information directly and the reward is computed based on 2D information only. However, we believe that the 3D reasoning is performed implicitly by the RL. Because in order to achieve the goal, the RL will need to find the 3D arrangement of objects whose 2D projection matches the 2D demonstrations. Simply moving the object in 2D will never enable the peg inside the box and thus even the 2D goal will not be achieved. RL can figure out how to manipulate the objects in 3D because it has 3 degrees of freedom.
> >
> > Having said that, GraphIRL wouldn’t work for tasks which require precise 3D manipulations such as performing a medical procedure however in such cases, 2D images also might not be sufficient and 3D positions of objects would be required for any method to be successful. We acknowledge this concern in the limitation section of our revision.
> >
> >
> > **Q**: On Line 250, it is claimed that the proposed method outperforms the oracle method. This conclusion is not fully proven by the data as it stands. [..], and after this point it is unclear which method will perform better, since neither curve looks like it has reached convergence.
> >
> > **A**: We agree with the reviewer. We will retract this statement. For completeness, we re-run RL training for Peg in Box for 1.5 Million steps. We find that SAC (Environment reward) outperforms GraphIRL for this task (0.800 vs 0.616). Please see the success rate curve
> > [here](https://drive.google.com/file/d/1c32LtF9C_isBTnUchJACDuzoAfQLysRR/view?usp=sharing).
> >
> > We will re-run other baselines and add the results in the final version. We note that GraphIRL significantly outperforms vision based baselines for this task.
> >
> > **Q**: The limitations section does not explore the limitations of the method in detail, such as those mentioned in the review.
> >
> >
> > **A**: We will add a more detailed discussion of limitations in our revision.

---

> > > ### Author Response · Authors · 2022-08-26
> > > **Authors' response to Reviewer A9wW [part 3/3]**
> > >
> > > **Issues**
> > >
> > > **Q**: In Eq (4), why is g not just the final frame, but an aggregation across frames?
> > >
> > > **A**: In Eq (4), g is computed over the final frame and the aggregation is performed across all training demonstrations. Thus, the representative goal frame is the average of embeddings for final frames for all demonstrations.
> > >
> > > **Q**: Another important advantage of third-person videos, such as those used in this approach, compared to demonstrations involving a robot is the availability of in-the-wild data.
> > >
> > > **A**: We thank the reviewer for pointing this out. We will add this in the final version.
> > >
> > > **Q**: The description of TCC was quite concise, but it may be worth spending more space on this to explain it in more detail as it is quite key to the paper. The diagram from the original TCC paper conveys the intuition well.
> > >
> > > **A**: We will expand the description on TCC in the final version.
> > >
> > > **Q**: In Eq (3), lowercase s is not defined, it seems. Probably it should be the same as uppercase S.
> > >
> > > **A**: That’s correct, this will be fixed in the final version, thanks for pointing it out.
> > >
> > > **Q**: It may be helpful to emphasise that SAC is an oracle method in the real world experiments, in that it has access to reward function information
> > >
> > > **A**: Thanks, we will add this comment in the “Baselines” section. We will replace the word “SAC” with “Environment Reward” in the paper for more clarity.

---

### Official Review · Reviewer_J5yc · 2022-08-01

**Originality:** Excellent
**Technical Quality:** Excellent
**Clarity Of Presentation:** Excellent
**Impact:** 4

**Recommendation:**

Strong Accept: I recommend accepting the paper and will argue for my recommendation even if other reviewers hold a different opinion.

**Summary:**

This paper introduces an inverse reinforcement learning approach for learning from diverse third-person videos. The key component of the method is representing the frames of videos with graphs, which capture scene semantics and object interactions while leaving out visual details such as textures. They generate the graphs by running a pre-trained object detector then computing geometric relationships between each pair of detected objects; then, they apply temporal cycle consistency-based self-supervised learning on the graph representations, which yields an embedding space from which they can generate a dense reward signal given an observation and a goal. They show much improved performance with their method on a simulated sweeping task and three robotic manipulation tasks on a real robot.

**Issues:**

no specific issues, addressed in weaknesses

**Quality Of The Limitations Section:**

Limitations are addressed clearly

**Reviewer Expertise:**

4: The reviewer is confident but not absolutely certain that the evaluation is correct

**Robotics Focus:**

Sufficient demonstration on hardware

**Strengths And Weaknesses:**

Strengths:
1. Graph-based method is intuitive, well-explained, and shown to be very effective.
2. Experiments are thorough.
3. Writing is very clear.

Weaknesses:
1. The selection of architecture for processing the graph and building the graph could be better motivated. For instance, were other graph convolutional architectures considered? Why use a fully-connected graph? etc.
2. Regarding the graph representation, it is unclear exactly how exactly it differs from prior work in object-centric representations.

**Summary Of Recommendation:**

The paper shows that graph representations can be used to greatly improve the performance of inverse RL when learning from diverse data. The method is elegant, clearly explained, and highly effective, and the experiments are thorough, with results shown on a real robot.

---

> ### Author Response · Authors · 2022-08-26
> **Authors' response to Reviewer J5yc**
>
> We thank the reviewer for their thoughtful comments. Below, we address the individual comments:
>
> **Q**: The selection of architecture for processing the graph and building the graph could be better motivated. For instance, were other graph convolutional architectures considered? Why use a fully-connected graph? etc.
>
> **A**: We chose interaction networks based architecture because it's relatively simple and has been applied in the context of modeling object dynamics which is closely related to the problem considered in this work. Having said that, graph convolutional networks are also a potentially valid choice for this problem. A more systematic study would be required to understand the pros and cons of both architectures in this context whereas our work focuses more on showing the benefits of using our proposed graph abstraction compared to directly using images.
>
> The fully-connected graph is used because it allows us to model interactions of all objects with each other. Thus, this makes the least assumption about the problem. We want the learning algorithm to learn the relations between objects from data without having to construct manually designed task graphs.
>
> We will add more motivation regarding our design choices in the final version.
>
> **Q**: Regarding the graph representation, it is unclear exactly how exactly it differs from prior work in object-centric representations.
>
> **A**: The representation used for objects varies depending on the problem. For instance, [5, 37] use object position and velocity of the object for modeling object dynamics. Works on model predictive control [41, 42, 43] use the aforementioned properties along with other information such as object mass, inertia etc.  On the other hand, we only rely on 2D location, pairwise distances and object identities and learn representations by constructing graph abstraction over this high level information. We do not extract dynamics related properties since those differ across embodiments and aren’t readily available when using human demonstrations. Whereas, it is straightforward to construct the aforementioned representation using an off-the-shelf object detector.

---

### Author Response · Authors · 2022-08-28
**Revised Manuscript**

We would like to once again thank the reviewers and meta reviewer for their thorough and positive responses. Based on your feedback, we have made the following changes to our paper:

* Added a more detailed limitations section.
* Added additional experiment on the X-MAGICAL benchmark to clarify performance concerns.
* Added results on Peg in Box with 1.5 million steps to clarify concerns regarding convergence.
* Replaced the ablation on demonstrations that we had previously with the more comprehensive data ablations that we did during the rebuttal. The ablations clearly demonstrate the data efficiency of GraphIRL.
* Added additional references as per suggestion of Reviewer 77Z4.
* Added explanation regarding higher performance of learned reward as compared to the environment reward.
* Improved Figure 2 i.e. added notations used in the text to facilitate understanding.
* Added more explanation regarding the design choice of using fully-connected graphs.
* Provided details on the detector used for obtaining object bounding boxes.

The changes made have been highlighted in blue for clarity.

---

### Meta-Review · Area_Chair_YYwB · 2022-08-12

**Recommendation:** Accept (Oral)
**Confidence:** 5

**Metareview:**

This paper proposes an interesting approach to learning rewards from a diverse set of videos and has received some strong reviews. The approach is original, and the paper well-motivated and well-written. The experimental section is thorough with good results, and includes experiments on a real robot. Using a graph-based method for IRL is novel and could have significant impact. However, there were a few weaknesses found by the reviewers. In particular, more justification for the choice of architecture and visual features could improve the paper. How would different choices impact performance? Along with this, a more in-depth discussion of the limitations of the approach would greatly improve the paper. The authors are encouraged to address these concerns along with other concerns raised by the reviewers.

=========

Thank you to the authors for the detailed response and updated experiments.

---

> ### Author Response · Authors · 2022-08-26
> **Authors' response to Area Chair YYwB**
>
> We appreciate the thoughtful and positive responses from you and the reviewers regarding our approach and results. Below we address individual comments:
>
> **Q**: In particular, more justification for the choice of architecture and visual features could improve the paper. How would different choices impact performance?
>
> **A**: **Choice of architecture**: We chose interaction networks based architecture because it's relatively simple and has been applied in the context of modeling object dynamics which is closely related to the problem considered in this work. Moreover, we model the objects using a  fully-connected graph because it allows us to model interactions of all objects with each other. Thus, this makes the least assumption about the problem. We want the learning algorithm to learn the relations between objects from data without having to construct manually designed task graphs.
>
> **Visual features**: We choose to work on graph abstraction instead of visual features because it improves the generalization capabilities of the learned reward. As we show in our experiments on 4 tasks, the domain gap between pre-training data and reinforcement learning environment leads to substantial decrease in performance for baseline which rely on visual features. Therefore, directly adding visual features to GraphIRL would make it susceptible to failure when there's a significant domain gap.  Therefore, a more principled approach for fusing visual features into the proposed graph abstraction such that robustness to domain gap is preserved would be required and such an approach could lead to improved performance. This is an exciting future direction but is beyond the scope of our current work.
>
> **Q**:  Along with this, a more in-depth discussion of the limitations of the approach would greatly improve the paper.
>
> **A**: We thank the reviewers and AC for bringing up the limitations of our work. We have answered the reviewers' comments in the following responses and added a new summary of limitations in our revision.